Quantifying the land and population risk of sewage spills overland using a fine-scale, DEM-based GIS model

McDaniel Emma L. 1 2 emcdaniel10@gsu.edu
Atkinson Samuel F. 3 4
Tiwari Chetan 1 2 5
1 Center for Disaster Informatics and Computational Epidemiology, Georgia State University , Atlanta, Georgia , United States of America
2 Department of Computer Science, Georgia State University , Atlanta, Georgia , United States of America
3 Department of Biological Sciences, University of North Texas , Denton, Texas , United States of America
4 Advanced Environmental Research Institute, University of North Texas , Denton, Texas , United States of America
5 Department of Geosciences, Georgia State University , Atlanta, Georgia , United States of America
Oehlmann Jörg
Electronic publication date: 2023 Nov 20
Publication date: 2023
Volume: 11
Electronic Location ID: e16429
Received 2023 Jun 1; Accepted 2023 Oct 18
Copyright: © 2023 McDaniel et al.
Copyright year: 2023
Copyright holder: McDaniel et al.
License: This is an open access article distributed under the terms of the Creative Commons Attribution License, which permits unrestricted use, distribution, reproduction and adaptation in any medium and for any purpose provided that it is properly attributed. For attribution, the original author(s), title, publication source (PeerJ) and either DOI or URL of the article must be cited.
License URL: https://creativecommons.org/licenses/by/4.0/

Keywords: Quantifying risk, Sewage spills, Overland spill model, Land risk, Population risk, GIS model, Spatially explicit model

Funding: The authors received no funding for this work.

==============================
Accidental releases of untreated sewage into the environment, known as sewage spills, may cause adverse gastrointestinal stress to exposed populations, especially in young, elderly, or immune-compromised individuals. In addition to human pathogens, untreated sewage contains high levels of micropollutants, organic matter, nitrogen, and phosphorus, potentially resulting in aquatic ecosystem impacts such as algal blooms, depleted oxygen, and fish kills in spill-impacted waterways. Our Geographic Information System (GIS) model, Spill Footprint Exposure Risk (SFER) integrates fine-scale elevation data (1/3 arc-second) with flowpath tracing methods to estimate the expected overland pathways of sewage spills and the locations where they are likely to pool. The SFER model can be integrated with secondary measures tailored to the unique needs of decision-makers so they can assess spatially potential exposure risk. To illustrate avenues to assess risk, we developed risk measures for land and population health. The land risk of sewage spills is calculated for subwatershed regions by computing the proportion of the subwatershed’s area that is affected by one modeled footprint. The population health risk is assessed by computing the estimated number of individuals who are within the modeled footprint using fine-scale (90 square meters) population estimates data from LandScan USA. In the results, with a focus on the Atlanta metropolitan region, potential strategies to combine these risk measures with the SFER model are outlined to identify specific areas for intervention.

Introduction

Inadequate capacity, aging infrastructure, or severe weather events can lead to systematic patterns of accidental overland spills of liquid contaminants thereby causing disproportionate levels of risk to certain populations and areas in a region. Strategies to identify high-risk areas that can be targeted for proactive mitigation interventions would be a useful tool. We have developed a methodology called Spill Footprint Exposure Risk (SFER) that uses spill locations and high-resolution digital elevation models (DEMs) to create spill footprints that can then be used to calculate risk. For example, the SFER model can be used to assess environmental or population risk to overland spills. These footprints can be used at the scale of a single spill location to identify risk areas for one spill, but can also be aggregated to assess risk at larger regional scales.

A common accidental overland spill involves raw sewage discharges onto land, sometimes referred to as sanitary sewer overflows (SSOs). These spills prompt reactive responses to protect land and people from detrimental impacts. Typically, responses tend to be limited and site-specific. Proactive strategies such as using historical data to define high-risk spill areas and corresponding at-risk populations could provide opportunities to develop plans for improved regional resilience. These high-risk sewage spill areas can be prioritized for maintenance and thus alleviate some of the future risks of spills. Examples of modeling efforts that explicitly focus on overland spills for asphalt and oil are seen in works by Guo (2006), Mahdi, Shakibaeinia & Dibike (2020), and Chuang, Thorleifson & Barba (2021).

Other modeling approaches include the use of hydrologic models to estimate their pathways and impacts on a given region. Shrestha et al. (2005) combined flow tracelines and sampling to assess the water quality impact of spills of various sizes in Los Angeles, California. Quijano et al. (2017) developed 3D simulations of hydrodynamics and water quality in the Chicago Area Waterway System after two storms. Nguyen, Peche & Venohr (2021) provide a review of exfiltration modeling from sewer systems to groundwater. Salihu et al. (2022) provide a comprehensive literature review on modeling techniques for deteriorating sewer networks and sewer pipelines focusing on identifying the key factors that can be used for modeling efforts. Yang & Wang (2010) conduct a literature review of models used to assess water quality in various watersheds in Europe for the implementation of the European Water Framework Directive.

Other researchers have developed spatially explicit approaches using Geographic Information Systems (GIS) and other tools to model spills and their impacts. Examples include: Foster & McDonald (2000) in the British Uplands, Guillen, Rainey & Morin (2004) in the Gulf of Mexico, and Zhou (2022) in Puget Sound Basin. Other modeling approaches have focused on mapping spills to assist in developing response strategies after such a spill occurred see work by Morgenstern (2003) for oil spills, and Jiang et al. (2012) for chemical spills.

The United States (US) Environmental Protection Agency (EPA) estimates there are at least 23,000–75,000 SSOs per year, which does not include sewage backups (U.S. Environmental Protection Agency, 2022). In the state of Georgia alone, between May 2020 and January 2023, there were a reported 1,909 spills totaling approximately 287 million liters (76 million gallons) of sewage spilled onto land (Georgia Environmental Protection Agency, 2023). These spills were recorded with a variety of primary causes; the top three reasons were: 19.77% by debris, 18.87% by wet weather, and 17.33% by grease; see Table 1 for the full breakdown of causes. The primary cause “wet weather” is not defined by the curator of the dataset. It is unclear if this means they are combined sewer overflows (CSOs), which are often caused by inundation of water into the sewerage system during heavy rain, or another reason. CSOs are a commonly studied type of overflow/sewage spill. However, the SFER model is designed to create a footprint of overland spills and does not directly apply to CSOs which discharge into waterbodies.

Table 1 Breakdown of primary causes of sewage spills in Georgia dataset between May 2020–January 2023 (Georgia Environmental Protection Agency, 2023).

Spill primary cause	Number of records	Percentage*	
Debris	227	19.77	
Wet weather	220	18.97	
Grease	201	17.33	
Pipe failure	151	13.01	
Equipment failure	89	7.67	
Roots	46	3.97	
Rags	45	3.88	
Other	41	3.53	
Hydraulic overload	39	3.42	
Unknown	35	3.02	
Power failure	22	1.9	
3rd party contractor	16	1.38	
Vandalism	10	0.86	
Unspecified	8	0.69	
Contractor	6	0.52	
Operator errors	3	0.26	
Debris, Roots	1	0.09	
Note:

* Rounded to two decimal points.

Exposure to pathogens and contaminants caused by sewage spills can result in significant human health and environmental impacts. Other researchers have conducted comprehensive reviews on such impacts of wastewater. In a recent review, Sojobi & Zayed (2022) provide a detailed assessment of exposure to sewage via sewer overflows, including a summary of pathogens, methods of contamination, and diseases caused by pathogens. Xagoraraki, Yin & Svambayev (2014) review research on viruses in water systems and with a focus on wastewater; these viruses can cause a variety of health problems to animals and humans. Owolabi, Mohandes & Zayed (2022) identify the environmental impacts that sewage overflows have had on soil, air, quality of water, and businesses and structures. In Luo et al. (2014), they describe untreated micropollutants (e.g., pharmaceuticals, personal care products, industrial chemicals, and pesticides) in sewage water and review studies that found that these micropollutants can cause antibiotic resistance in microorganisms, threaten wildlife, and jeopardize drinking water quality. Although the SFER model does not explicitly measure dose response or contaminants in the spill itself, it provides a mechanism to estimate potential exposure to sewage spills.

Multiple studies examined the association between sewage contamination and/or public health utilizing field sampling approaches. Drayna et al. (2010) and Olds et al. (2018) assess the connection between weather events and water quality in Wisconsin. Miller, Ebelt & Levy (2022) and Rothenberg et al. (2023) identify if there are associations between combined sewage overflows and gastrointestinal disease reports in Atlanta, Georgia, and South Carolina respectively. Korajkic, Brownell & Harwood (2011) compare samples of water quality before and after a repair of sewer mains relocation of portable restrooms at a Florida gulf coast beach. Aulenbach, Henley & Hopkins (2023) conduct a correlation study to examine associations between water quality and place characteristics of the area from which samples were taken. Hagage, Madani & Elbeih (2022) utilize inverse distance weight to estimate the water quality of the groundwater based on 32 groundwater samples of the Quaternary aquifer in the Akhmim district of Upper Egypt; their goal was to identify the suitability of water across the region.

Understanding the spatial patterns of sewage spills on dry land can help address systemic vulnerabilities, thereby reducing the potential impacts of such spills. Climate change is expected to increase the risk of exposure to gastrointestinal disease-causing pathogens (Von Bonsdorff & Maunula, 2013) as well as worsen existing vulnerabilities in the built and natural environments through increasing stress on existing infrastructure (Hughes et al., 2021). There is much research on models that take into account the hydraulic modeling (Shrestha et al., 2005; Quijano et al., 2017; Nguyen, Peche & Venohr, 2021; Salihu et al., 2022; Yang & Wang, 2010) of spills into water bodies, but there is yet a simple GIS-based model for sewage spills occurring on dry land that only requires elevation and spill origin. These studies and conclusions suggest a need to develop a data-driven tool that allows decision-makers to identify areas of high risk and thus apply appropriate interventions as it relates to dry land. The SFER methodology generates modeled spill footprints that can be applied to different regional contexts and can be used in conjunction with secondary datasets, such as population density, to identify where interventions are most needed or useful from a proactive perspective. In this article, we apply this model to spatial contexts of both subwatershed and census tract regions and develop land and population risk indices that reflect levels of severity in exposure.

Methods

The SFER methodology was developed to model overland pathways of sewage spills, see Fig. 1 for the methodology flow chart. The model requires two inputs: the location of a spill(s) and surface elevation data. These datasets are widely available across the United States and other countries. The model is sensitive to the accuracy of the spill location information and the resolution of the elevation raster. Uncertainty in the geocode of the spill location may result in different spill footprints because of the possible influence of elevation differences. Ideally, Global Positioning System’s (GPS) recorded locations should be utilized as the spill start points. The resolution of the elevation raster can be increased or decreased depending on the scale at which the SFER model is desired. Over the last decade, DEMs have increased in accuracy because of the inclusion of LIDAR cloud points. In particular, the USGS 3D Elevation Program (3DEP) rasters and USGS’ quality control measures (Stoker & Miller, 2022) have increased significantly thus allowing for such an elevation based model such as SFER to be beneficial. While the input data of the SFER model reported in the Results section utilizes a 3DEP 1/3 arc second elevation raster, there are 1-m rasters available for select portions of the U.S. that if used would result in finer-scale models. Unlike other similar modeling projects, the input DEM was not made depressionless, therefore the traceline will end when a cell only has neighbor cells that flow into it. We chose this method because we did not intend to analyze the fate of the spilled material, but instead wanted to find the first point of pooling because that would represent the initial area of sewage accumulation and highest contaminant concentration. Farrar et al. (2005) use a similar technique in modeling the flow of oil spills overland; however, they include additional parameters such as flow velocity and elevation to create the size of their trace line; they also do not include accumulation zones in their model.

Figure 1 Methodology flow chart to create the SFER model.

All processing of the datasets was performed within a PostgreSQL (13.9) database (PostgreSQL Global Development Group, 2022), with the PostGIS extension (3.3.2) (PostGIS Development Team, 2022). Minimal data processing was completed using Esri’s ArcGIS Pro 3.0.3 (Esri, 2022). In the appendices, we include all the preprocessing steps to implement the SFER model. Some data were converted into appropriate spatial data types and resolutions as necessary for compatible spatial format/support in order to apply functions and other techniques.

Model design

The spill model is composed of three components that represent where sewage from a determined spill location may flow across land and accumulate in areas using an elevation raster; see Fig. 2. The first component (purple) is the area surrounding the geocoded spill address location; we call this an accumulation zone. The second component is the flow traceline (green). The third component is an accumulation zone surrounding the end point of the traceline (orange).

Figure 2 SFER is composed of three polygons, the start point accumulation zone (purple), the traceline (green), and the end point accumulation zone (orange).

Accumulation zones

The accumulation zones (components 1 and 3) are polygons surrounding the start and end points. The start point is the geocoded position of where the spill is indicated as starting. The end point is the point where the traceline (described below) terminates. We create accumulation zones to represent possible pooling areas around these start/end points. The process for creating both these zones is the same for start and end point.

First, we calculate the potential accumulation zone using a dynamically sized buffer based on the spill quantity, see Eq. (1). The size, si, is dependent on quantity, qi of the ith spill, Q is the set of all quantities. The formula is a proportional equation to calculate the buffer in relation to the max quantity (max(Q)) of all the spills in the dataset. We chose a maximum buffer size of 50 m and the size of the specific spill’s quantity ( qi) is calculated proportionally. We chose to take the log of the spill quantity in order to mitigate outliers in spill quantities that were found in our dataset. An illustration of this calculated possible buffer size can be seen in Fig. 3, as the dark red circle. This buffer size can change depending on the use-case of the SFER model, and the outlier normalization is not necessary and the potential buffer size can be increased or decreased depending on the perceptions of the type of spill (e.g., necessary safety factor) by a decision-maker.

Figure 3 The green line is the traceline.

The red dot is the end point of the traceline. The white are elevation points. The dark red polygon is the potential accumulation zone buffer. The orange polygon is the estimated accumulation zone.

(1) si=log(qi)log(max(Q))50

After the potential accumulation buffer is created, the areas of elevation that are below the start/end point are aggregated. First, the elevation value of either the start or end point is compared to the elevation value of all the other points in the potential accumulation zone buffer. In Fig. 3, the focus is on the end point accumulation zone, but the process would be the same for the start point. The dark red point in the middle of the figure is the end point and the number 314 is the elevation value in meters above sea level.

To more easily compare the elevation of the end point to surrounding cells in the elevation raster, the elevation raster is converted into points using a raster to point GIS function. A buffer of 4.5 m was created around these points to approximate the area that would be covered by the original 1/3 arc second (approximately 5-m resolution) raster. Simply using the point location instead of buffered points would result in an underestimation of the accumulation zone. See Appendix A.III for further information on building this layer. In Fig. 3, the elevation points are depicted as semi-translucent white points and their corresponding labels are the elevation raster cells values. The elevation points (white points) within the potential accumulation buffer region (dark red polygon) are compared with the start/end point (red point) and aggregated using the PostGIS (2023a) ST_Collect and then turned into a polygon (orange geometry) using the PostGIS (2023b) ST_ConcaveHull functions. In Fig. 3, the orange polygon encapsulates the elevation values that are less than 314 m and represents the accumulation zone.

Tracelines

The second component of the SFER model is called the flow traceline, represented by the green line in Fig. 1. First, we must create a D8 flow direction raster derived from the elevation raster. In our case, we utilized the 1/3 arc second elevation tiles for all of Georgia (U.S. Geological Survey, 2023) were utilized as input into the Flow Direction tool in ArcGIS Pro (Esri, 2023a). This tool has the parameter “Force all edge cells to flow outward,” and was set to “Normal.” This means that at the edge cells of the raster, the flow will go outward (Jenson & Domingue, 1988). For our implementation for the state of Georgia, this parameter was chosen because we did not find any spills that would “pour” over the side of the elevation. However, this choice is strictly up to those utilizing this model, and their use-case and should be made appropriately.

From the D8 raster, tracelines that follow the directions on the D8 raster cells, beginning at an inputted start point until all the cells are flowing into itself were generated utilizing the ArcGIS Pro tool called Flow Path Tracing (Arc Hydro) (Esri, 2023b), with the input of (1) start points, and (2) the D8 raster. This tool allows for custom D8 rasters, unlike the similar Trace Downstream tool (Esri, 2023c). With the tracelines generated, a 15 m buffer was used to represent a worst case path. This value could be modified to represent other scenarios based on decision makers knowledge of regional circumstances.

When combined, the two accumulation zones and the traceline polygons composed the SFER model. The SFER model was developed in such a way that it is fairly simple to implement and results in a worst case footprint of any kind of overland spill. In this model, if a spill of a specific volume occurs in the same location no matter when it occurs, it will have the same spill footprint, because of the elevation based strategy. If the spill is larger in volume, the spill may have larger accumulation zone. The Postgres queries to create the model are available at https://osf.io/hs6g9/.

Assessment of risk

Using the SFER model, exposure risk to populations and to land can be assessed. Two metrics within the domain of sewage spills that could be applicable for decision-makers to assess were developed. First, the risk that historical sewage spills have posed to the environment, and second the public health risk of human community exposure can be examined.

To evaluate public health risk, a metric utilizing an estimated population exposed by the spills within each census tract was generated. Census tracts are regions as defined by the U.S. Census Bureau (U.S. Census Bureau, 2021a). Census tracts were chosen because of their ability to be joined with other demographic information like age or income level. The estimated ratio of people exposed to a spill to the people living in the census tract indicates which census tracts are at higher risk of exposure to its inhabitants. The population estimated by the SFER model is calculated using the LandScan USA Night 90-m raster (Weber et al., 2021). The LandScan raster was converted into points and a 35 m buffer was added; see Appendix A.III for description of how this layer is built and Fig. 4 for a visual representation of the calculation of this ratio. In this figure, the dotted circles with numbers are the LandScan points with the value of the number of persons estimated to live within the corresponding raster cell. The orange polygons are the SFER models. Where these spill models intersect the LandScan points, these points have a bolded outline. The intersecting points with the SFER models are summed, and that value represents the estimated persons exposed to a spill. If there are spills that occur in the same location, the SFER model is first unioned, so the population values impacted are not double counted.

Figure 4 Spill footprints (orange) within a census tract (purple) that intersect LandScan polygons (dotted or bolded lined circles) with a base map of buildings and roads.

The numbers within the LandScan polygons indicate population counts.

The population risk metric can be seen in Eq. (2). The union of n spill geometries A1,A2,A3,...An that intersect with a census tract, ct, results in a geometry that is the union of the spills within that ct. This unioned geometry represents the area that may be been exposed to sewage within the census tract, and thus with another intersection of Landscan Raster, L, it is possible to sum the population that may be been exposed to the spill. This metric can be applied across all census tracts in the region being analyzed.

(2) pRisk=sum(((A1∪A2∪A3,...An)∩ct)∩L)population (ct)

The second metric to calculate risk is a ratio that indicates risk of the land area in a subwatershed region that may be contaminated by a sewage spill originating from a previously reported spill. These subwatershed regions are from the National Hydrology Dataset Plus (U.S. Geological Survey, 2022). These are regions that have been designated as flowing into a single stream segment. In Fig. 5, there are two watershed subregions (blue polygons designated with a and b) and the spills (orange polygons) that occurred within the regions. The total area of subwatershed a is 1,398,358 m2, and the one spill that occurred within it is 3,708 m2. For subwatershed b, the area is 630,274 m2, and of the six spills within this region (five occurred at the same location with quantity differences), their unioned area is 12,186 m2.

Figure 5 SFER models (orange) within each subwatershed region (blue) with the labels of the area m2 for each.

This land risk metric is shown in Eq. (3). The union of n spill geometries A1,A2,A3,...An that intersect with the subwatershed region, sw, results in a geometry that is the union of the spills within that sw. The resulting geometry represents the area of land that may have been exposed to the spill within the subwatershed region. The land area of the modeled spill footprint, over the entire subwatershed region area represents land risk.

(3) lRisk=area((A1∪A2∪A3,...An)∩sw)area(sw)

For both of these metrics, when spill footprints overlap, the union function described in the numerator is used to ensure that neither the population counts (Eq. (2)) nor the land areas (Eq. (3)) of the spill model are counted more than once. This was chosen to highlight areas where the risk is high irrespective of the number of times a spill has occurred at a given location. It is entirely possible to instead use the sum of the spill geometries’ indicating chronic spills, which is a risk on which some decision-makers may wish to focus.

Results and Discussion

We have developed a fine-scale, spatially explicit model, called Spill Footprint Exposure Risk (SFER), that can be used to estimate the potential risk to land and nearby populations by creating a “footprint” of a spill. Using this SFER model, we have developed metrics to identify spills that could have a higher impact on land and/or population health. An important feature of SFER is that historical data on the locations of previous spills can be used as an input to the model, allowing decision makers to examine where proactive planning could result in reduced risk and harm from events that could re-occur if preventative actions are not implemented.

Application of SFER model

To illustrate the utility of the SFER model, it was applied to sewage spill data reported by the Environmental Protection Agency (EPA) of the State of Georgia, USA for the 14-month period between December 2020 and January 2023 (Georgia Environmental Protection Agency, 2023). Georgia reported 1,909 spills, releasing a total of approximately 287 million liters of untreated sewage during this time window. Among the data reported for each spill, the address of spill release and the estimated volume of untreated sewage were provided.

To use these within our model, each spill location was first geocoded using the ArcGIS Online Geocoder service (see Appendix A.II for details). Of the total spills, SFER modeling generated spill footprints for 1,477 of these spills (about 78%). The 432 dropped records were due to geocoding errors and inconclusive quantity of spill amounts.

Of the 1,969 Census Tracts in Georgia, 768 experienced sewage spills in the 14-month window. A total population of 3,099,925 people are reported to live with the affected census tracts. Fine-scale estimates using the LandScan USA Night 90-m rasters of potential population exposed within these census tracts is estimated at 32,760. Of the total subwatershed areas containing a spill zone the overall subwatershed area sums to 161,063 hectares. The total area of the SFER models within this time window was 1,104 hectares. On average, based on the SFER model, 0.53% of the area of an affected subwatershed is impacted by a spill zone. In the maps and figures that follow, we only display census tracts and subwatersheds that have at least one spill incidence within the region.

We have created an interactive map with the layers for the land and population risk across all of Georgia, which can be accessed at elmcdaniel.com/GA_USA_2023_SFERMap. Also, the layers are readily accessible for download within our project repository (https://osf.io/hs6g9/). This map is composed of three layers: subwatershed regions ( lRisk), census tracts ( pRisk) for night, and census tracts ( pRisk) for day. The interactive map is classified using the quantile method with five classes for each of the layers. The lighter colors indicate lower values of the pRisk or lRisk metrics. There are few regions outside of urban areas that have recorded spills. This could be representative of sewage infrastructure differences between urban and non-urban environments or lack of reporting of these types of sewage spills. The difference in perceived impact of sewage spills between the census tracts and subwatershed region map is a result of the relative area covered by spatial units representing each domain. Census tracts must contain a certain population count, and thus in areas with lower population density, census tracts are larger. In areas with less population density, utilizing census tracts as the pRisk region may be less valuable, instead, it would be more beneficial to use finer scale administrative boundaries such as city/place.

Maps of population and land exposure may be used to inform different aspects of public policy making. Decision-makers may use the SFER models to prioritize areas within their jurisdictions where interventions may be needed. We focused on Atlanta as our jurisdiction of interest to illustrate the use of our methodology. We did this for two reasons: (1) sewage spills are of particular concern in the City of Atlanta in Georgia, which historically has been in non-compliance with requirements on regulating Combined Sewer Overflows (CSOs) and Seperate Sewer Overflows (SSOs) (U.S. District Court, Northern District of Georgia, Atlanta Division, 1998, 1999); and (2) the City of Atlanta has a department called Atlanta Watershed Management which is in charge of the sewage system for the city, and thus this could be a planning process they would undertake.

Quantification of risk

Decision-makers that are interested in assessing the risk of exposure to populations may find that it is beneficial to utilize census tracts, estimated population exposed via LandScan data, and our population risk metric as we have described. See Fig. 6A of the population exposure risk score in a choloropleth map that is classified using the quantile method (equal number of tracts per color). Central downtown Atlanta does not have high risk, which can could be attributed to the use of night time population data (in Appendix B, Fig. B1 is the corresponding with the Daytime Raster utilized). Higher risk in Fig. 6A is delegated to the outer rings of the city, where there are more residences and therefore more nighttime population. There are three clusters of adjacent census tracts in the northwest, northeast, and southeast that are within the highest quintile of risk, which may represent areas that decision-makers may choose to prioritize. The high exposure risk that is quantified with the pRisk metric in this case may indicate systemic problems such as aged infrastructure.

Figure 6 (A) Choropleth map of census tracts by population risk metric using night Landscan raster; (B) top quintile of population risk map by population under 5; (C) choropleth map of subwatershed regions by land risk metric; (D) top quintile of land risk metric by percentage of imperviousness.

Further, it could be beneficial to prioritize regions that contain a higher number of persons vulnerable to gastrointestinal diseases, like persons under the age of 5. This can be done by joining the census tracts to data available in either the Decennial Census or American Community Surveys (ACS), another benefit for utilizing this regional boundary. Figure 6B is a map that takes the top quintile of Fig. 6A, and overlays the counts of persons under 5 years old from ACS 2021 (U.S. Census Bureau, 2021b). In this case, decision-makers could target the western and southern clusters of the city as they have the highest counts of populations under 5 years old. Other demographic parameters of interest in ACS could be used to assess other indicators of vulnerability such as estimated counts of persons above 65 years old, under the poverty line, or medical insurance coverage.

For those interested in assessing environmental risk, they may wish to combine information from the lRisk map with other spatial information such as the percentage of impervious surface area in each subwatershed. In Fig. 6D, the mean imperviousness score was calculated for the top quintile of the subwatersheds in Fig. 6B using the Zonal Statistics as Table tool (Esri, 2023d) from the NLCD 2019 Percent Developed Imperviousness raster (U.S. Geological Survey, 2019). While subwatersheds in the central core of Atlanta contain urban landuse, most are also located in larger subwatersheds. As expected, the highest risk areas are primarily smaller watershed regions with higher percentages of impervious surfaces, and are scattered throughout the city. Impervious surfaces are often used as a surrogate measure of the amount of urbanization, and Melchiorri et al. (2018) have shown that rapidly urbanizing landscapes can adversely affect the natural resources which human populations rely upon. Impervious surfaces can concentrate and funnel overland flow, adversely affecting numerous environmental systems (e.g., nutrient cycling, stream temperature, flood attenuation), especially contaminant filtration and water purification (National Research Council, 2002). Multiple studies have shown that strategies applied to rapidly urbanizing areas can be used protect high quality environmental resources (Atkinson, Hunter & English, 2010) or restore degraded environmental resources (Atkinson & Lake, 2020).

Prioritizing locations

Public health planners and watershed managers working together can simultaneously explore both demographic data that inform public health decisions and watershed management decisions to concentrate limited resources in the most efficient combination. One method to do this is illustrated in the top figure in Fig. 7. This map of Atlanta contains pins indicating where the top quintiles for population risk (Fig. 6A) and land risk (Fig. 6C) intersect. This could be a method for decision-makers to prioritize broad locations of concern. In particular, there are a collection of five pins in the northeast portion of Atlanta that could be prioritized. Additionally, these locations may be qualitatively examined for place context using web mapping tools (e.g., Google Maps, Earth, or Streetview) and mobility data (e.g., SafeGraph). The bottom of Fig. 7 displays the street view (Google Maps, 2022) of a park located adjacent to a community farm where multiple spills were found to have occurred totaling an estimated 662,123 liters of sewage spilled. If decision-makers are aware of certain high-risk areas, they can qualitatively assess these areas in relation to the SFER model.

Figure 7 Top: intersection of the top quintiles of population and land risks for the City of Atlanta; bottom: qualitative assessment example with spill quantity information and Google Streetview of a high-risk area.

Limitations

The SFER model is sensitive to the accuracy and resolution of the input datasets used. Inaccuracies in spill locations and elevation data will impact the footprints generated and the derived measures of environmental or population risk. In our study, spill locations were determined by geocoding the reported addresses of where spills were recorded. The exact latitude and longitude of the source of spills as well as the use of higher-resolution DEM can result in a more robust footprint. Further, the use of aggregated population data at the census tract level may introduce spatial biases in population-based assessments of risk. For example, census tracts classified as high risk due to the presence of selected population vulnerabilities may be inaccurate if those populations are clustered in areas with no spills. While high-resolution data on population estimates like LandScan USA may be used to identify overall population exposure at fine-spatial scales, information on specific population characteristics at that scale are not available.

Estimates of land risk using SFER represent relatively simple overland flow models as a function of spill volume and surface elevation. Other environmental parameters may either exacerbate or attenuate the actual environmental risk. For example, if the spill flows through a high-quality wetland, the contaminates may be degraded much more quickly as compared to the spill accumulating along roadways. Likewise, a spill that accumulates in a recreational area such as a community baseball complex or a dog-park may expose more people than census or LandScan USA population estimates would indicate. Including accurate landuse/landcover data into the type of proactive planning that SFER suggests would improve the plans that are developed. The SFER model is limited to use for dry land spills and treats any input location of spills as dry land spills. It also does not account for differences in the surfaces. The incorporation of modeling spills using SFER Model with hydraulic components was out of scope for this model as the focus is on dry land spills.

Conclusion

Our focus of sewage spills in our study is of environmental and public health concern as sewage spills can impact a region’s population and land health. The SFER model and corresponding methodology to assess risk of overland spills has the intention of making available quantifiable metrics that allow decision-makers to prioritize at-risk areas, which can help uncover and systemic disparities in a region.

The SFER model can be used to model other types of overland spills in different contexts. For example, the model could be applied to hazardous spills from industrial facilities or to model areas of greatest risk from chemical spills from train derailments. The SFER model can be combined with other attributes that represent population health or the environment to develop additional measures of risk. Further, the model can be applied to selective areas to prioritize spill locations that are near parks, outdoor recreations, medical facilities as reported by OpenStreetMap (OpenStreetMap Foundation, 2023) or Homeland Infrastructure Foundation-Level Data (Homeland Infrastructure Foundation-Level Data (HIFLD), 2021).

Supplemental Information

Supplemental Information 1 Data preprocessing, Day Landscan figure with pRisk for Atlanta, and notes on figure creation.

Click here for additional data file.

Additional Information and Declarations

Competing Interests

Author Contributions

Data Availability

The authors declare that they have no competing interests.

Emma L. McDaniel conceived and designed the experiments, performed the experiments, analyzed the data, prepared figures and/or tables, authored or reviewed drafts of the article, and approved the final draft.

Samuel F. Atkinson conceived and designed the experiments, analyzed the data, authored or reviewed drafts of the article, and approved the final draft.

Chetan Tiwari conceived and designed the experiments, analyzed the data, authored or reviewed drafts of the article, and approved the final draft.

The following information was supplied regarding data availability:

The data and code are available at OSF: McDaniel, Emma L, and Chetan Tiwari. 2023. “Quantifying the Land and Population Risk of Sewage Spills Overland Using a Fine-Scale, DEM-Based GIS Model.” OSF. October 12. DOI 10.17605/OSF.IO/HS6G9.

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
