# Peer review of "Quantifying the land and population risk of sewage spills overland using a fine-scale, DEM-based GIS model"

_PeerJ, doi:10.7717/peerj.16429_

## Round 0.1 · original submission · Major Revisions

I have to inform you that your submission was assigned to me as academic editor (AE) because Mykola Karabiniuk, who originally served as AE, is unavailable.

Both reviewers highlight the importance of your submission but especially reviewer 1 identified a number of issues that have to be addressed before your manuscript can be accepted. These include the amendment of results and conclusion(s) in your abstract and clarification of your underlying concept of risk with a specification of resulting health and environmental implications. In addition, it should be clarified (or excluded) whether the model can also be used for the discharge of untreated sewage into surface waters, as is regularly the case with storm water overflow discharges after rain events to protect the sewage treatment plants from the loss of their microbial community in the biological treatment.

Reviewer 2 proposes an altered title. This suggestion and also the literature suggestions to be considered for the introduction are not mandatory points for me.

I would also suggest that you address the threat to human health and the environment from chemical pollution other than "organic matter, nitrogen, and phosphorus" in the manuscript, specifically micropollutants, for which there are numerous studies (e.g. https://dx.doi.org/10.1016/j.watres.2015.09.023).

I look forward to your revised manuscript.

Reviewer 1 ·

Basic reporting

Abstract
Very well-structured abstract – introduced the problem, objectives, and the methodology of the research, although it does not say anything about the results obtained or the conclusion. An abstract should inform about the summary results and conclusion of a completed study.

Introduction
Need to restructure sentence in Lines 58 – 59
Need to restructure sentence in Lines 68 – 69
You may want to elaborate a bit on the point of discussion in Lines 66 – 73 i.e. the Public health impacts of raw sewage disposal from combined sewer overflows or spillage from SSOs. What are the public health implications? What circumstances cause these spillages? Are there situations where POTWs (Public Owned Treatment Works) release raw sewage into water bodies? Some useful references for you could include: 10.1128/AEM.01203-07, https://doi.org/10.1016/j.marpolbul.2022.114212
It would be beneficial to have expatiated more on the causes of sewage spills. You mentioned infrastructure conditions however in the case of CSOs, they combine both municipal/industrial sewage with stormwater and are sometimes intentionally discharged into water bodies to prevent overwhelming Treatment Plants.

Experimental design

Methodology
Not many comments here, the basis for the model is well established however, does this model only capture sewage spills on dry land surfaces? If it captures sewage spills and dumping into water bodies, what role will water flow rates, site-specific conditions, and seasonality play in the model parameters? If on land, how do new accounts for uncertainties caused by precipitation and runoff which may alter the trace line and accumulation end zone predicted by the model?

Validity of the findings

Results & Discussion
Lines 213 – 228 should not be in this section. It should be part of the methodology and a tabular format for the information would be better
Punctuation needed in Line 229
Line 234 – “On average, 0.53% of the area of an affected sub-watershed is impacted by a spill zone.” Is this a fact garnered from your study results? If not, provide a reference.
The risk route for the human population is not well discussed. While the sewage path and exposure area are determined and mapped out, it is important to discuss the tangible ‘hows’ of how people are at risk. Is it from water consumption from water bodies, or seepage into groundwater and thus private wells (maybe map out private wells exposed within the sewage spill zone) or is it cropland or through vector transfer?

Additional comments

This study addresses a very important topic - sewage spills and the associated risk that human and biota ecosystems are exposed to as a consequence of the spills. The objective of the study is clear – to develop a geographical-based model for quantifying the associated risks from sewage spillage for life within the spillage zone.
However, the study does not sufficiently address the risk aspect in its introduction and discussion. The study needs to sufficiently answer questions on how humans are exposed to risk from sewage spills, and what is the consequence for the aquatic biota.
The study does not also clarify the variations that may exist with the type of surface where spillage occurs – ideally, the model addresses a dry land spillage scenario but does not sufficiently indicate if hydrology and gradient are important factors to consider nor does it address possible and very common spillage that occurs into water bodies.

Reviewer 2 ·

Basic reporting

Dear Dr. Mykola Karabiniuk,
I apologize for the delay in my review of your manuscript. I have now had a chance to read it carefully, and I have some comments that I believe will improve the paper.
Overall, the paper is well-written and informative. However, there are a few areas that could be improved.
Title: The title is clear and informative, but it could be shorter and more concise. For example, you could write "Sewage Spill Risk Assessment using a GIS Model" or "GIS Modeling of Sewage Spill Exposure and Impacts."
Abstract: The abstract is well-structured and covers the main aspects of the paper. However, it could be improved by following some guidelines for scientific writing. Specifically, you should clearly state the usefulness of the study and formulate a key message that summarizes the main finding or implication of the study.
Introduction: The introduction is well-written and provides a good overview of the topic. However, it could be improved by adding a paragraph on the role of remote sensing and geographic information systems in environmental studies related to water. You could use the following references to help you write this paragraph:
Groundwater deterioration in Akhmim District, Upper Egypt: a remote sensing and GIS investigation approach (Salwa F Elbeih, Ahmed A Madani, Mohammed Hagage)
Quaternary groundwater aquifer suitability for drinking in Akhmim, Upper Egypt: an assessment using water quality index and GIS techniques( Mohammed Hagage, Ahmed A Madani, Salwa F Elbeih)
Random Forest and Logistic Regression algorithms for prediction of groundwater contamination using ammonia concentration(Ahmed Madani, Mohammed Hagage, Salwa F Elbeih)
Methods: The methods section of the manuscript provides a clear and concise overview of the research methodology used.
I recommend that you include a flowchart in the methods section that visually represents the research methodology. This will help the reader to understand the steps that were involved in the study.
I recommend that you rewrite the conclusion to focus more on the results and their importance.
In addition to these specific comments, I would also recommend that you review the paper carefully for grammar and spelling errors.
I believe that these changes will make the paper stronger and more likely to be accepted for publication. Thank you for your time and effort in writing this paper.
Sincerely,
Mammed Hagage

Experimental design

no comment

Validity of the findings

no comment

Additional comments

no comment

---

## Round 0.2 · accepted · Accept

Thank you for the thorough and comprehensive revision of the manuscript. I hereby certify that you have adequately taken into account all of the reviewers' comments, as demonstrated by my own assessment of your revised manuscript. Based on my assessment as an Academic Editor, your manuscript is now ready for publication.